# Population dynamics and biological feasibility of sustainable harvesting as a conservation strategy for tropical and temperate freshwater turtles

Angga Rachmansah[1¤], Darren Norris[2,3]*, James P. Gibbs[1]

**1** Department of Environmental and Forest Biology, State University of New York College of Environmental Science and Forestry, Syracuse, New York, United States of America, **2** Ecology and Conservation of Amazonian Vertebrates Research Group, Federal University of Amapá, Macapá, Amapá, Brazil, **3** Postgraduate Programme in Tropical Biodiversity, Federal University of Amapá, Macapá, Amapá, Brazil

¤ Current address: Fauna Flora International – Indonesia Programme, Jakarta, Indonesia
* dnorris75@gmail.com

**Data Availability Statement:** All relevant data are within the paper and its Supporting Information files.

## Abstract

### Background

Conservation strategies are urgently needed for tropical turtles that are increasingly threatened by unsustainable exploitation. Studies conducted exclusively in temperate zones have revealed that typical turtle life history traits (including delayed sexual maturity and high adult survivorship) make sustainable harvest programs an unviable strategy for turtle conservation. However, most turtles are tropical in distribution and the tropics have higher, more constant and more extended ambient temperature regimes that, in general, are more favorable for population growth.

### Methods

To estimate the capacity of temperate and tropical turtles to sustain harvest, we synthesized life-history traits from 165 predominantly freshwater turtle species in 12 families (Carettochelydae, Chelidae, Chelydridae, Dermatemydidae, Emydidae, Geoemydidae, Kinosternidae, Pelomedusidae, Platysternidae, Podocnemididae, Staurotypidae and Trionychidae). The influence of climate variables and latitude on turtle life-history traits (clutch size, clutch frequency, age at sexual maturity, and annual adult survival) were examined using Generalized Additive Models. The biological feasibility of sustainable harvest in temperate and tropical species was evaluated using a sensitivity analysis of population growth rates obtained from stage-structured matrix population models.

### Results

Turtles at low latitudes (tropical zones) exhibit smaller clutch sizes, higher clutch frequency, and earlier age at sexual maturity than those at high latitudes (temperate zones). Adult survival increased weakly with latitude and declined significantly with increasing bioclimatic

**Funding:** This work was supported by the United States Agency for International Development (AID-OAA-A11-00012). The funders had no role in study design, data collection and analysis, decision to publish, or preparation of the manuscript.

**Competing interests:** The authors have declared that no competing interests exist.

temperature (mean temperature of warmest quarter). A modeling synthesis of these data indicates that the interplay of life-history traits does not create higher harvest opportunity in adults of tropical species. Yet, we found potential for sustainable exploitation of eggs in tropical species.

## Conclusions

Sustainable harvest as a conservation strategy for tropical turtles appears to be as biologically problematic as in temperature zones and likely only possible if the focus is on limited harvest of eggs. Further studies are urgently needed to understand how the predicted population surplus in early life stages can be most effectively incorporated into conservation programs for tropical turtles.

## Introduction

Vertebrate animals are important for human welfare and wellbeing [1–3], particularly as food, medicine, and cultural uses by rural and aboriginal communities [3–6]. Freshwater turtles are a good example—they are frequently targeted for both subsistence and commercial harvest, primarily by local communities that live in the vicinity of river and wetlands [7–9]. High biomass [10, 11], ease of capture, and extended survival with minimal care in captivity make freshwater turtles a focus for harvest [7–9].

Unsustainable harvesting is recognized as one of the major factors driving global freshwater turtle decline [12–15]. Over 40% of turtle species are endangered as a result of overexploitation [13, 15, 16]. Although turtles are harvested for various purposes (e.g. pets, medicine, and curios), the most heavy use of turtles is for food [7, 16, 17]. Large adult turtles [18–21] and eggs [18] are usually the primary target of harvesting, because these life stages are the most valuable for food [7, 8, 16] and the easiest life stages to encounter. The greatest harvesting pressure occurs in tropical areas [7, 8], where the most freshwater turtles occur [22, 23]. For many local people in these areas, turtle meat and eggs are not only important as sources of protein and lipid, but also support them economically [7, 16, 24]. Yet, unsustainable exploitation in tropical areas can also lead to regional population collapse and as a consequence create pressures in other regions of the world [25].

Sustainable harvesting programs have been widely promoted as a strategy for wildlife conservation [26, 27]. Moreover, active involvement of local people in these sustainable harvest programs generally creates better outcomes for conserving wildlife [27, 28]. However, this conservation strategy is assumed not viable for turtle conservation [7, 29]. A corpus of research on the topic has revealed that turtles are poor candidates for any sustainable use program [30–32]. In general, turtles exhibit delayed sexual maturity, high adult survivorship, low fecundity, and long life span [30–35]. This combination of life-history traits limits their ability to compensate for additive adult mortality from harvesting [9, 29, 33, 35, 36].

It is notable, however, that virtually all research on sustainability of harvest as a conservation strategy for turtles has been conducted in temperate zones. Variation in life-history traits occurs within and between turtle species that inhabit different environments [33, 37–41]. Variation in clutch size [37, 42], clutch frequency [34], growth rate, and age at sexual maturity [37] in relation to latitude have been observed in turtles. The interplay of these different life-history traits has been suggested to create more opportunity to harvest turtles sustainably, at least in one tropical freshwater species in Northern Australia [19, 43]. Earlier age at sexual maturity, higher fecundity, and faster growth rates in this tropical freshwater turtle compared to other

turtles [43] may allow their populations to be harvested at 20% annual harvest rate [19]. This elevated harvest rate challenges the generality of the widely held assumption that sustainable harvest programs are biological infeasible for freshwater turtles. Indeed, considering that sustainable harvest research is based almost entirely on temperate zone species, the biological feasibility of sustainable harvest should be reassessed given the challenges of conserving turtles in rapidly developing tropical regions where most turtle diversity occurs [9, 13].

In this study, we investigated global patterns of life-history traits (clutch size, clutch frequency, age at sexual maturity, and adult survival) in freshwater turtles using published data and contrasted them between freshwater turtle species from temperate and tropical regions. We then developed a population projection model to estimate the capacity of freshwater turtle species from temperate and tropical regions to sustain harvest. The primary goal of this study was to evaluate the hypothesis that freshwater turtle species from tropical and temperate regions have the same, widely speculated incapacity to absorb additive mortality caused by population harvest [30, 31, 36].

## Materials and methods

### Data collection

Life-history traits of freshwater turtle species were quantified along with locality of each report (latitude and longitude) from the published literature. We used keywords "life history", "clutch size", "clutch frequency", "reproduction", "age at sexual maturity", "survival", "growth", "natural history", and "turtle" to explore the published literature as indexed in the databases of EBSCO, Google Scholar, and Web of Science. Marine turtles (Cheloniidae and Dermochelyidae) and tortoises (Testudinidae) were excluded from the results. Although some families (e.g. Emydidae) include a few predominantly terrestrial turtle species (e.g. *Terrapene carolina*), as our analysis is general across taxonomic families, hereafter we refer to all as "freshwater turtles" to distinguish them from marine turtles or tortoises. The mean, median, or range (midpoint calculated and used in < 1% of cases) values of reproductive parameters (clutch size, clutch frequency), demographic parameters (age at sexual maturity, annual adult survival rate), and morphological characters (carapace length) were extracted from each report acquired. Annual adult survival values were also checked and confirmed against those available for 15 freshwater turtle species in an online demographic database COMADRE [44] (version 3.0.0, accessed 2 September 2019 http://www.comadre-db.org/Data/Comadre).

When the exact coordinates of locality were not described, we estimated location from the nearest locality described in a given report. The coordinates of each turtle life-history report were also combined with Global Biodiversity Information Facility (GBIF) records (accessed via GBIF.org on 2019-01-13) and published data [39] to establish species distribution across four latitudinal classes: Temperate (species with latitudinal median and range within temperate zone), Temperate-tropical ("Temp-trop", species with latitudinal median within temperate and range overlapping tropical zone/s), Tropical-temperate ("Trop-temp", species with latitudinal median within tropical and range overlapping temperate zone/s), Tropical (species with latitudinal median and range within tropical zone). The tropics of Capricorn and Cancer (latitude -23.5˚, 23.5˚, respectively) were used to define geographic limits of temperate and tropical zones.

Two bioclimatic variables relevant to freshwater turtle biology, Mean Temperature of Warmest Quarter (bio10, ˚C) and Precipitation of Driest Quarter (bio17, mm) were obtained from WorldClim—Global Climate Data (5-arc ≈ 10 km resolution, www.worldclim.org, [45]) and matched to the coordinates of each turtle life-history report using functions available in the R [46] package raster [47]. These bioclimatic variables were selected as proxies to represent

the metabolic, physiological and behavioral differences that freshwater turtles have developed to survive in regions that are not ideal for these temperature and water dependent species [10, 22, 33, 34, 39–42, 48]. Both bioclimatic variables were only weakly correlated with latitude (Spearmans correlation 0.40 and 0.04 for bio10 and bio17 respectively) and were therefore included to represent temperature and rainfall patterns distinct to those most strongly associated with latitudinal gradients.

## Statistical analysis

We used Generalized Additive Models (GAMs, [49, 50]) to examine the influence of climate variables and latitude on freshwater turtle life-history traits (clutch size, clutch frequency, age at sexual maturity, and annual adult survival). We treated each freshwater turtle species as a replicate in this analysis (obtaining median life-history values within species for species with $n > 1$ reports) to avoid the pitfalls of pseudoreplication associated with treating individual reports as replicates. Because comparative life-history studies are not independent from phylogenetic relationships among turtles, which can lead to phylogenetic bias on inference and trait value estimation, we treated taxonomic family as a random effect (penalized smoothed regression term) [51, 52] based on the Turtles of the World Checklist (8[th] edition, [53]). In addition, we used carapace length (ln-transformed) as a parametric term to control for its well-established influence on life-history traits [34, 37, 39, 42].

A total of four models were developed for each life-history trait: latitude as a continuous variable included as a parametric term, latitude as a categorical variable with four classes (Temperate, Temp-Trop, Trop-Temp and Tropical), and two bioclimatic variables (Mean Temperature of Warmest Quarter and Precipitation of Driest Quarter) included as parametric terms. All four model variables were only weakly correlated with carapace length (all pairwise Pearson correlations $< 0.25$, S1 File) so could be reliably included in the GAM analysis [50]. All life-history trait estimates were ln-transformed, except for adult survival (arcsine transformed). The mgcv package [49] was used to perform the GAM analysis in R (www.r-project. org, [46]). Akaike Information Criterion corrected for small sample sizes (AICc) that measures fit versus complexity of a model was used to select "best" models based on lowest AICc [54, 55].

## Modelling synthesis

To evaluate whether freshwater turtles from tropical and temperate zones have comparable capacities to absorb additive mortality caused by population harvest, we implemented a density-independent, stage structured "Lefkovitch" matrix population model [35, 56, 57]. We chose to retain this relatively simple model, without including nonlinear effects (e.g. density-dependence, environmental stochasticity) as although some turtles are very well-studied, the majority of species typically have many demographic and life-history parameters that are unknown or highly uncertain. The sparsity of original data available for both turtle demography and human impacts better support a stage-based approach [58]. We therefore consider a stage-structured parameterization as most appropriate for our objective to compare population dynamics of tropical and temperate turtles. The stage-structured model is also commonly used in turtle population dynamics modelling, as age in most turtle species (typically long-lived, iteroparous and mobile) is often difficult to determine [19, 35]. The model consisted of egg, juvenile, and adult stages (Fig 1) projected with a stable-stage distribution (with an initial population of 1000, allocated in proportions of 0.544, 0.401, 0.055 to egg, juvenile and adult stages respectively).

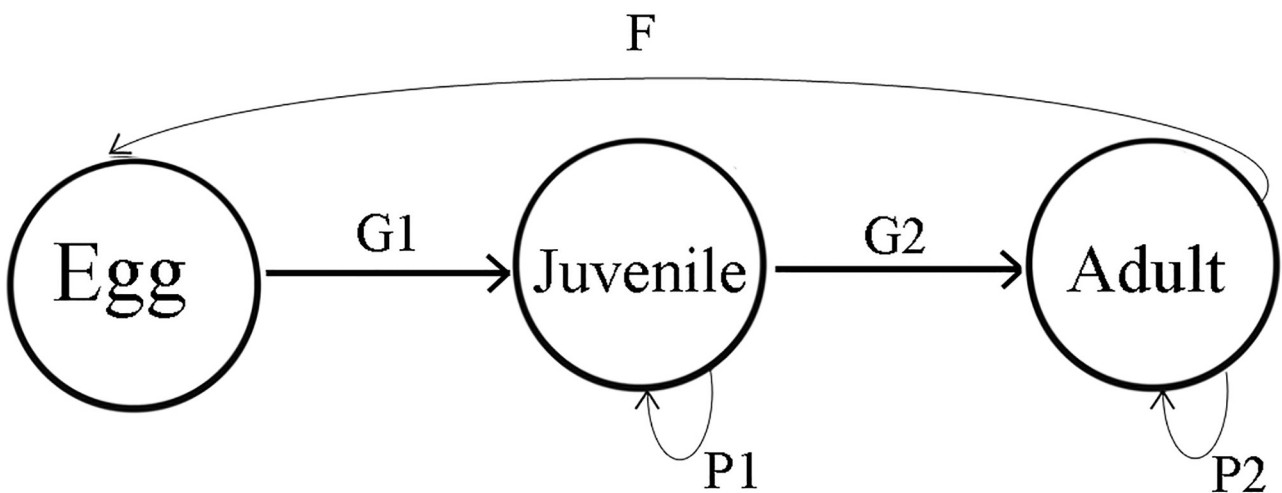

**Fig 1. Conceptual diagram of population dynamics of freshwater turtles used for construction of a stage structure matrix model to estimate capacity for sustainable harvest in freshwater turtles.**

The discrete stage based lifecycle (Fig 1) can be presented as a population projection matrix "A" as follows:

$$A = \begin{bmatrix} 0 & 0 & F \\ G_1 & P_1 & 0 \\ 0 & G_2 & P_2 \end{bmatrix}$$

where $P$ is the annual probability of surviving and remaining in the same stage, $G$ is the annual probability of surviving and growing into next stage, and $F$ is the annual fecundity. These parameters were estimated using the following equations [58]:

$$P = \frac{(1 - p_i^{d_i - 1})}{(1 - p_i^{d_i})} p_i \tag{1}$$

$$G = \frac{p_i^{d_i}(1 - p_i)}{1 - p_i^{d_i}} \tag{2}$$

where $p_i$ is the annual survival probability of $i$ stage and $d_i$ is the duration of $i$ stage. These equations assume asymptotic growth rate equals 1, and have been referred to as "stationary age-within-stage structure" models [59]. Annual fecundity ($F$) was estimated by multiplying clutch size with clutch frequency. The model was based on female fraction only; thus half of all eggs produced was assumed to be female [30, 31]. The stable distributions of individuals amongst stage classes and intrinsic rate of population growth ($r$) were determined with functions available in the R [46] packages "popdemo" [60] and "popbio" [61].

Median values of clutch size, clutch frequency, age at sexual maturity and adult survival derived from the GAM predictions was used as input for this stage-structured model. Due to the sparsity of records for some traits (e.g. adult survival) predictions were aggregated across two latitudinal classes (temperate and tropical) to compare the intrinsic rate of population growth ($r$) between stages and latitude. Predictions for each trait were obtained from a final GAM model that included all variables in a 95% confidence subset of models [54]. This confidence set was obtained by summing the Akaike weights of the set of all candidate models

ordered by Akaike weight from largest to smallest until a sum of $\geq 0.95$ was obtained ([54] pp. 169, 176–177). We estimated the annual survival probability of juvenile stage as 13% less than the annual survival probability of adult stage [62]. Due to lack of available nest / hatchling survival data the annual survival probability of egg stage for all turtle species was set at 0.2 [30, 31, 33].

Elasticity analysis was used to identify stages that should be the focus of management effort and that contribute most to fitness [35, 63, 64]. Elasticities (proportional change) differ from sensitivities (absolute change) given a change in the matrix parameter [35, 64]. By calculating elasticities it is therefore possible to compare i) the relative effects of proportional change in one or more life-history stages and ii) proportional changes in values (e.g. fecundity and survival) which are on different scales [35, 56]. Elasticities were compared from the stage structured matrix population models parameterized using both the median observed and median predicted values.

To simulate the impact of harvest on populations of our generalized tropical and temperate freshwater turtles, we performed a sensitivity analysis by varying each demographic parameter systematically while holding all other parameters constant [30, 31]. In addition, we performed Jackknife randomizations [65] drawing deviates (n = 500 iterations) for each model parameter from the distribution (95% value range) of species level values observed in the literature (S1 Table) for these variables to estimate confidence intervals around the estimated intrinsic rates of growth of temperate and tropical species in sensitivity analysis.

## Results

A total of 461 reports of life-history traits were obtained from 165 species (63% of living freshwater turtle species) among 12 taxonomic families (Fig 2, S1 Table). The data once aggregated (Table 1) represent: 84 species from 7 families in the temperate zone (Temperate and Temperate-Tropical classes) and 81 species from 12 families in the tropical zone (Tropical-Temperate and Tropical classes). Sixty percent of these studies were from temperate areas, with most of these (73%) from North America (Fig 2). Forty percent of these data were from tropical areas, with most of these (36%) from Asia. Only 12 of these life-history trait reports (including 5 tropical and 3 temperate species) were from captive breeding situations while the remainder were from wild populations.

Latitude as continuous variable significantly influenced all life-history traits, except adult survival (Fig 3, Table 2, Table 3). Indeed, latitude was the most informative variable for clutch size, clutch frequency and age at sexual maturity (Table 3). Natural logarithm of clutch size ($\beta$ = 0.13; $P < 0.001$) and age at sexual maturity ($\beta$ = 0.06; $P < 0.01$) were positively related to latitude, whereas natural logarithm of clutch frequency ($\beta$ = -0.09; $P < 0.05$) exhibited a negative relationship with latitude (Fig 3, Table 2). When latitude was treated as categorical variable, only the natural logarithm of clutch size was significantly related to latitudinal zones, such that Tropical ($\beta$ = -0.21; $P < 0.001$) and Tropical-Temperate ($\beta$ = -0.13; $P < 0.05$) species had reduced clutch size relative to temperate species (Table 2).

Of the two bioclimatic variables assessed, only bioclimatic temperature (Mean Temperature of Warmest Quarter) was a contributor to life-history variation (Table 2) and was also the most informative variable for adult survival (Table 3). The bioclimatic temperature models were included in the 95% confidence set for all life-history traits, except clutch size (Table 3). Natural logarithm of age at sexual maturity ($\beta$ = -0.06; $P < 0.01$) and arcsine adult survival ($\beta$ = -0.08; $P < 0.05$) were both negatively related to Mean Temperature of Warmest Quarter (Fig 4).

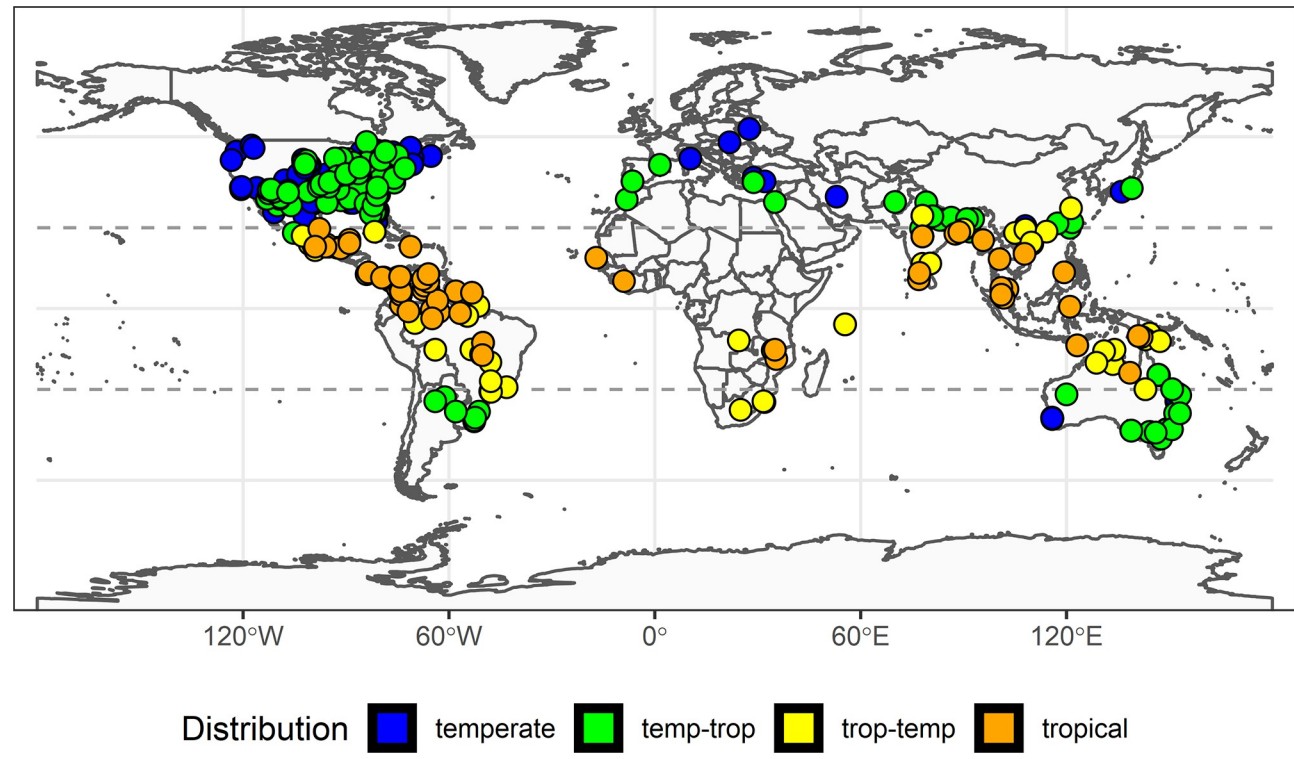

**Fig 2. Distribution of freshwater turtle studies.** Geographic distribution of data on freshwater turtle life-history traits obtained from the literature (S1 Table) to estimate capacity for sustainable harvest in freshwater turtles. Color of study locations represent the distribution of the study species across four latitudinal classes: Temperate (species with latitudinal median and range within temperate zone), "Temp-Trop" (species with latitudinal median within temperate and range overlapping tropical zone/s), "Trop-Temp" (species with latitudinal median within tropical and range overlapping temperate zone/s), Tropical (species with latitudinal median and range within tropical zone). Dashed horizontal lines show Tropic of Cancer and Tropic of Capricorn (latitude -23.5˚, 23.5˚, respectively). The background map was obtained from the 1:110m Natural Earth country and geographic lines maps (http://www.naturalearthdata.com).

Elasticity analysis revealed that the survival of adult females was by far (on average at least 2 times) more important than juvenile stages. In contrast, egg survival had the weakest effect on population multiplication rate. The trends in elasticities between stages were similar for both temperate and tropical species. Yet, on average both egg and juvenile survival tended to be

**Table 1. Demographic parameters in freshwater turtles.** Demographic parameters used in population modelling to estimate capacity for sustainable harvest in freshwater turtles. Estimates are median values derived from the scientific literature (S1 Table) and summarized based on the species distributions across four latitudinal classes. Values in parentheses are the number of species with data available and used to calculate medians.

| Distribution[a] | Families | Species | Lat[b] | Carapace Length | Clutch Size | Clutch Frequency | Age at sexual maturity | Fecundity |
|---|---|---|---|---|---|---|---|---|
| Temperate | 6 | 41 | 34.0 | 181.0 (41) | 8.4 (41) | 2.0 (35) | 8.7 (34) | 7.8 (35) |
| Temp-Trop | 6 | 43 | 29.1 | 221.7 (43) | 11.2 (43) | 1.7 (28) | 8.3 (18) | 6.6 (28) |
| Trop-Temp | 10 | 37 | 18.3 | 197.3 (37) | 6.1 (37) | 2.5 (20) | 6.5 (12) | 6.0 (20) |
| Tropical | 10 | 44 | 9.6 | 231.5 (44) | 7.3 (44) | 2.0 (19) | 9.0 (11) | 3.5 (19) |
| Overall | 12 | 165 | 23.1 | 208.0 (165) | 8.0 (165) | 2.0 (102) | 8.3 (75) | 6.3 (102) |

[a] Distribution of freshwater turtles in four latitudinal classes: Temperate (species latitudinal median and range within temperate zone), Temp-trop (species latitudinal median within temperate and range overlapping tropical zone/s), Trop-temp (species latitudinal median within tropical and range overlapping temperate zone/s), Tropical (species latitudinal median and range within tropical zone). This classification is unique for each species i.e. a species is only included in one class.

[b] Median latitude from species locations within each distribution class.

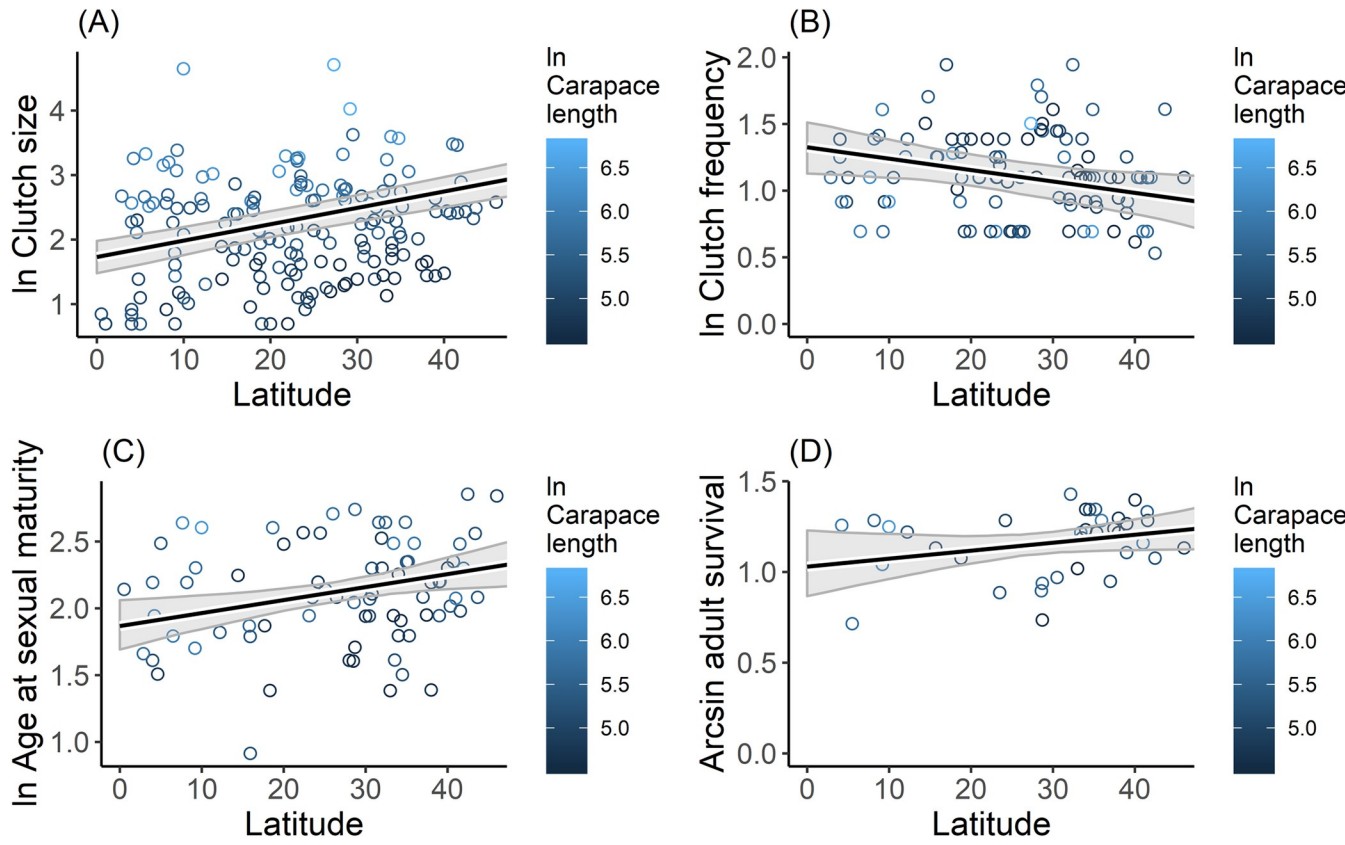

**Fig 3. Relationships between latitude and (A) clutch size, (B) clutch frequency, (C) age at sexual maturity, and (D) adult survival rate of freshwater turtles.** Points are the median species values obtained from the literature (S1 Table), colored representing ln carapace length values. Solid black line is the GAM prediction. Grey shaded polygons show 95% confidence bands around the prediction.

relatively more important in tropical compared with temperate species (Table 4). The sensitivity analysis performed to examine the impact of harvest on freshwater turtle populations revealed that adult and juvenile survival rates had dramatically more impact on intrinsic rate of population growth than egg survival rate and fecundity (Fig 5). Tropical freshwater turtle species exhibited a moderately higher intrinsic rate of growth than temperate freshwater turtle species (Fig 5). Although, fecundity tended to be less in tropical species (Table 1, Table 5), comparing the minimum values necessary to result in positive population growth with GAM predictions showed that fecundity could be reduced by 28% in tropical compared with only 12% in temperate species (Table 5). Survival rates were estimated to be reducible by 35% in eggs, 24% in juveniles, and 5% in adults for tropical species, and 15%, 16%, and 7%, respectively for temperate species, without causing negative population growth (Table 5). However, overlap in estimations of population growth in relation to survival rates was very broad between tropical and temperate turtle species (Fig 5, Table 5).

## Discussion

The capacity of any species to cope with additive mortality is determined by the interplay of its life-history traits [63, 66, 67]. Turtles are often declared to share integrated life-history traits [68] that make compensation for additive mortality associated with harvest infeasible [29]. Life-history traits of many organisms are related to variation in environment [69, 70], climate

**Table 2. Influence of climate variables and latitude on freshwater turtle life-history traits.** Generalized Additive Models were used to predict responses of four freshwater turtle life-history traits: clutch size, clutch frequency, age at sexual maturity and adult survival.

| Model | | ln clutch size (n = 165) | | | ln clutch frequency (n = 102) | | | ln age at sexual maturity (n = 75) | | | arcsine adult survival (n = 37) | | |
|---|---|---|---|---|---|---|---|---|---|---|---|---|---|
| **Continuous latitude** | | Est.[a] | SE | p | Est.[a] | SE | p | Est.[a] | SE | p | Est.[a] | SE | p |
| Intercept | | 0.78 | 0.05 | *** | 0.12 | 0.05 | * | 0.75 | 0.02 | *** | 0.12 | 0.03 | *** |
| Log carapace length | | 0.29 | 0.02 | *** | 0.01 | 0.03 | | 0.09 | 0.02 | *** | 0.01 | 0.03 | |
| Latitude | | 0.13 | 0.00 | *** | -0.09 | 0.04 | * | 0.06 | 0.02 | ** | 0.04 | 0.03 | |
| **Smooth** | | **Edf/ref** | **F** | **p** | **Edf/ref** | **F** | **p** | **Edf/ref** | **F** | **p** | **Edf/ref** | **F** | **p** |
| Family | | 8.1/11 | 6.48 | *** | 5.0 / 11 | 1.2 | * | 0.0 / 10 | 0.0 | | 0.0 / 7 | 0.0 | |
| R² ajust / Dev. Exp [b] | | 0.78 / 77.6% | | | 0.12 / 17.9% | | | 0.19 / 20.2 | | | 0.01 / 6.7 | | |
| **Categorical latitude** | | Est.[a] | SE | p | Est.[a] | SE | p | Est.[a] | SE | p | Est.[a] | SE | p |
| Intercept | | 0.85 | 0.06 | *** | 0.12 | 0.06 | . | 0.76 | 0.05 | *** | 0.10 | 0.06 | |
| Log carapace length | | 0.29 | 0.02 | *** | 0.01 | 0.03 | | 0.06 | 0.03 | * | -0.01 | 0.04 | |
| Latitude | temp-trop | 0.04 | 0.05 | | -0.06 | 0.08 | | 0.02 | 0.05 | | 0.08 | 0.06 | |
| | trop-temp | -0.13 | 0.05 | * | 0.06 | 0.09 | | -0.11 | 0.07 | | 0.07 | 0.09 | |
| | tropical | -0.21 | 0.05 | *** | 0.04 | 0.09 | | -0.02 | 0.08 | | -0.05 | 0.12 | |
| **Smooth** | | **Edf/ref** | **F** | **p** | **Edf/ref** | **F** | **p** | **Edf/ref** | **F** | **p** | **Edf/ref** | **F** | **p** |
| Family | | 7.9/11 | 6.27 | *** | 2.7 / 11 | 0.4 | | 4.7 / 10 | 0.9 | . | 2.5 / 7 | 0.6 | |
| R² ajust / Dev. Exp [b] | | 0.76 / 75.8% | | | 0.03 / 9.3% | | | 0.24 / 32.2% | | | 0.07 / 23.8% | | |
| **Bioclimate - temp** | | Est.[a] | SE | p | Est.[a] | SE | p | Est.[a] | SE | p | Est.[a] | SE | p |
| Intercept | | 0.76 | 0.05 | *** | 0.12 | 0.04 | ** | 0.72 | 0.04 | *** | 0.11 | 0.03 | ** |
| Log carapace length | | 0.27 | 0.02 | *** | 0.01 | 0.03 | | 0.07 | 0.03 | ** | 0.02 | 0.03 | |
| Temp. warm quarter (bio10) | | -0.01 | 0.02 | | 0.03 | 0.03 | | -0.06 | 0.02 | ** | -0.08 | 0.03 | * |
| **Smooth** | | **Edf/ref** | **F** | **p** | **Edf/ref** | **F** | **p** | **Edf/ref** | **F** | **p** | **Edf/ref** | **F** | **p** |
| Family | | 7.0 / 11 | 4.39 | *** | 3.4 / 11 | 0.6 | . | 5.4 / 10 | 1.2 | * | 0.0 / 7 | 0.0 | |
| R² ajust / Dev. Exp [b] | | 0.72 / 69.8% | | | 0.05 / 9.9% | | | 0.31 / 37.7% | | | 0.12 / 16.7% | | |
| **Bioclimate - rain** | | Est.[a] | SE | p | Est.[a] | SE | p | Est.[a] | SE | p | Est.[a] | SE | p |
| Intercept | | 0.76 | 0.05 | *** | 0.12 | 0.04 | ** | 0.73 | 0.05 | *** | 0.15 | 0.03 | *** |
| Log carapace length | | 0.27 | 0.02 | *** | 0.01 | 0.03 | | 0.06 | 0.03 | * | -0.01 | 0.03 | |
| Rain dry quarter (bio17) | | -0.03 | 0.02 | . | 0.02 | 0.03 | | 0.00 | 0.02 | | 0.02 | 0.04 | |
| **Smooth** | | **Edf/ref** | **F** | **p** | **Edf/ref** | **F** | **p** | **Edf/ref** | **F** | **p** | **Edf/ref** | **F** | **p** |
| Family | | 7.4/11 | 5.71 | *** | 2.5 / 11 | 0.3 | | 5.7 / 10 | 1.5 | * | 0.0 / 7 | 0.0 | |
| R² ajust / Dev. Exp [b] | | 0.72 / 70.5% | | | 0.02 / 7.0% | | | 0.24 / 31.6% | | | -0.04 / 1.4% | | |

Each model contained Family as a random effect (smooth GAM term specified with "re" basis) and body size (ln transformed carapace length) as a parametric term; Asterisks indicate significant level of estimated parameters (*** $P < 0.001$; ** $P < 0.01$; * $P < 0.05$; '.' $P < 0.1$).

[a] Standardized regression coefficient (obtained by dividing the centered response values by their standard deviations) and associated standard error (SE).

[b] Model adjusted r-squared and deviance explained (%)

[71] and their ecological interactions [63, 66, 67, 72, 73] and this study revealed that turtle life-history is strongly related to latitude and ambient temperature. Yet although these trends might suggest an increase in capacity of tropical freshwater turtles to absorb additional mortality due to anthropogenic sources than in temperate zone species, once integrated in a synthetic population model tropical species appear to be as unable to absorb additive mortality as are temperate zone species.

The positive relationship we observed between clutch size and latitude is consistent with earlier studies [37, 42]. Turtles that inhabit higher (temperate) latitudes, have larger clutch size than turtles that inhabit low (tropical) latitudes. Similar patterns have been observed in mammals [71] and birds [72, 74]. Tokolyi, Schmidt (71) and McNamara, Barta (72) suggest this

**Table 3. Freshwater turtle life-history model comparisons.** Comparisons of the Generalized Additive Models created for each life-history trait to estimate capacity for sustainable harvest in freshwater turtles. Models for each trait ordered by decreasing AICc (Akaike information criterion corrected for small sample sizes) values.

| Life-history trait | Model[a] | Dev. Exp | Loglik | BIC | AICc | Δ AICc | $W_i$ AICc[b] |
|---|---|---|---|---|---|---|---|
| Clutch size | | | | | | | |
| | Continuous latitude | 77.6 | -69.35 | 207.05 | 168.09 | 0.00 | 1.00 |
| | Categorical latitude | 75.8 | -80.04 | 237.57 | 193.80 | 25.71 | 0.00 |
| | Bioclimate - rain | 69.8 | -95.48 | 256.12 | 218.85 | 50.77 | 0.00 |
| | Bioclimate - temp | 70.5 | -98.96 | 261.27 | 224.98 | 56.90 | 0.00 |
| Clutch frequency | | | | | | | |
| | Continuous latitude | 17.9 | -14.42 | 81.83 | 54.94 | 0.00 | 0.93 |
| | Bioclimate - temp | 9.9 | -19.25 | 85.31 | 61.22 | 6.28 | 0.04 |
| | Bioclimate - rain | 7.0 | -21.01 | 83.69 | 62.00 | 9.79 | 0.03 |
| | Categorical latitude | 9.3 | -19.61 | 91.34 | 64.84 | 12.62 | 0.01 |
| Age at sexual maturity | | | | | | | |
| | Continuous latitude | 20.2 | -28.82 | 79.24 | 68.52 | 0.00 | 0.57 |
| | Bioclimate - temp | 37.7 | -19.46 | 92.49 | 69.13 | 0.62 | 0.42 |
| | Bioclimate - rain | 31.6 | -22.86 | 99.50 | 76.08 | 7.56 | 0.01 |
| | Categorical latitude | 32.2 | -22.62 | 104.54 | 79.42 | 10.91 | 0.00 |
| Adult survival | | | | | | | |
| | Bioclimate - temp | 16.7 | 14.16 | -10.39 | -16.31 | 0.00 | 0.85 |
| | Continuous latitude | 6.7 | 12.09 | -6.27 | -12.19 | 4.12 | 0.11 |
| | Bioclimate - rain | 1.4 | 11.09 | -4.26 | -10.18 | 6.13 | 0.04 |
| | Categorical latitude | 23.8 | 15.77 | 7.28 | 0.73 | 17.04 | 0.00 |

[a] Models used to predict natural history traits. Each model contained Family as a random effect (smooth term with "re" basis) and body size (log transformed carapace length) as a parametric (not smooth) effect. Continuous latitude included median latitude from all records (S1 Table). Categorical latitude included four latitudinal classes: Temperate (species with latitudinal median and range within temperate zone), "Temp-Trop" (species with latitudinal median within temperate and range overlapping tropical zone/s), "Trop-Temp" (species with latitudinal median within tropical and range overlapping temperate zone/s), Tropical (species with latitudinal median and range within tropical zone). Bioclimate—temp included Mean Temperature of Warmest Quarter (WorldClim: bio10). Bioclimate—rain included Precipitation of Driest Quarter (WorldClim: bio17). Coefficients for individual variables in all models are presented in Table 2.

[b] Akaike weights ($W_i$) from largest to smallest. Predictions for each trait were obtained using variables from the 95% confidence subset of models, obtained by first ordering all models in the set by decreasing Akaike weight ($W_i$), and then sequentially summing the model $W_i$'s in rank order.

pattern is related to climate variability. Iverson, Balgooyen (42) concluded that higher juvenile competition due to shorter time period for development along with higher egg mortality associated with winter and climate uncertainty that creates temporary periods of low competition may make it more advantageous for temperate turtle species to produce more offspring ("more eggs in one basket" [34]) as a "bet hedging" strategy to exploit temporary resources. In addition, temperate turtle species typically have small egg size to speed development as an adaptation to short incubation times in temperate zone [17, 42]. As such, our findings support the suggestion that temperature zone turtles may have evolved to produce smaller egg size with larger clutch size than tropical species [34].

Larger clutch size in temperate turtle species may also act as a mechanism to compensate for low nesting frequency [34, 42]. We found that clutch frequency was negatively related to latitude. The general model of the interaction of environmental factors and reproductive output in turtles [34] suggests that high latitudes yield short reproductive seasons for turtles, resulting in lower clutch frequency. In addition, timing of nesting in turtles is correlated with temperature [37, 75]. Because tropical zones have a more stable warmer temperature all year long, more opportunities are available for turtles to lay eggs than in the temperate zone.

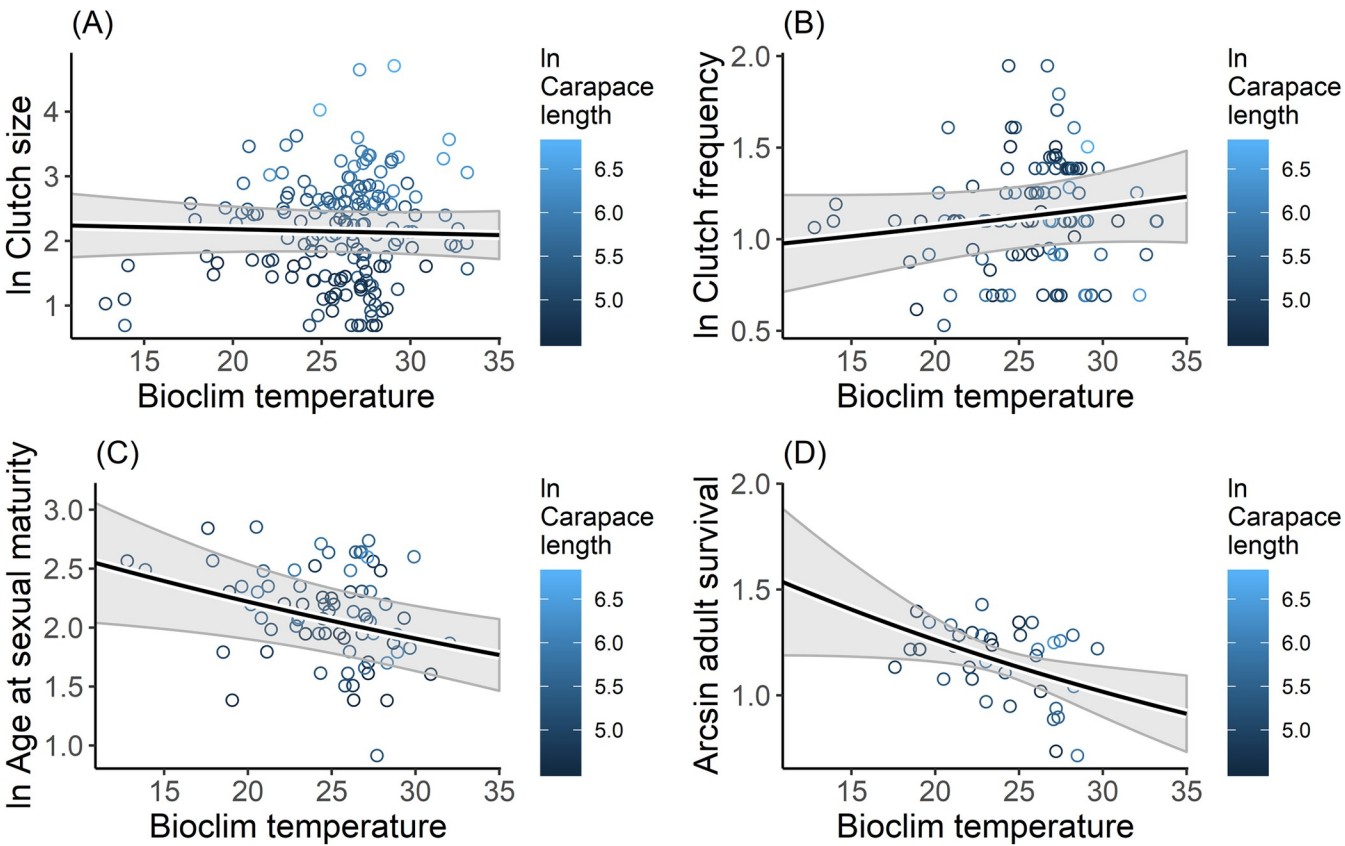

**Fig 4. Relationships between bioclimatic temperature (Mean Temperature of Warmest Quarter) and (A) clutch size, (B) clutch frequency, (C) age at sexual maturity, and (D) adult survival rate of freshwater turtles.** Points are the median species values obtained from the literature (S1 Table), colored representing ln carapace length values. Solid black line is the GAM prediction. Grey shaded polygons show 95% confidence bands around the prediction.

Additionally, clutch mass (number of eggs x egg size) and reproductive output (often estimated as relative clutch mass) can also vary with latitude [34, 42]. Further studies are necessary to examine how reproductive output correlates to differences in population growth rates, especially as egg size and reproductive output have been shown to be important predictors of age at sexual maturity [33, 34].

**Table 4. Elasticity values.** Elasticity values calculated from stage structured matrix population models for demographic parameters in temperate (Temp.) and tropical (Trop.) freshwater turtles. Observed elasticities were derived from median values from the scientific literature (S1 Table) and predicted values are from the 95% confidence set of GAM models (Table 3).

| Parameter | Observed | | Predicted | |
|---|---|---|---|---|
| | Temp. | Trop. | Temp. | Trop. |
| Egg survival | 0.090 | 0.110 | 0.076 | 0.082 |
| Juvenile $P_i$[a] | 0.183 | 0.252 | 0.208 | 0.216 |
| Juvenile $G_i$[b] | 0.090 | 0.110 | 0.076 | 0.082 |
| Adult survival | 0.546 | 0.419 | 0.564 | 0.537 |
| Annual fecundity[c] | 0.090 | 0.110 | 0.076 | 0.082 |

[a] Juvenile $P_i$ = probability of a juvenile surviving and remaining in the juvenile stage.

[b] Juvenile $G_i$ = probability of a juvenile surviving and "graduating" to the adult stage.

[c] Clutch size X number of clutches X breeding frequency

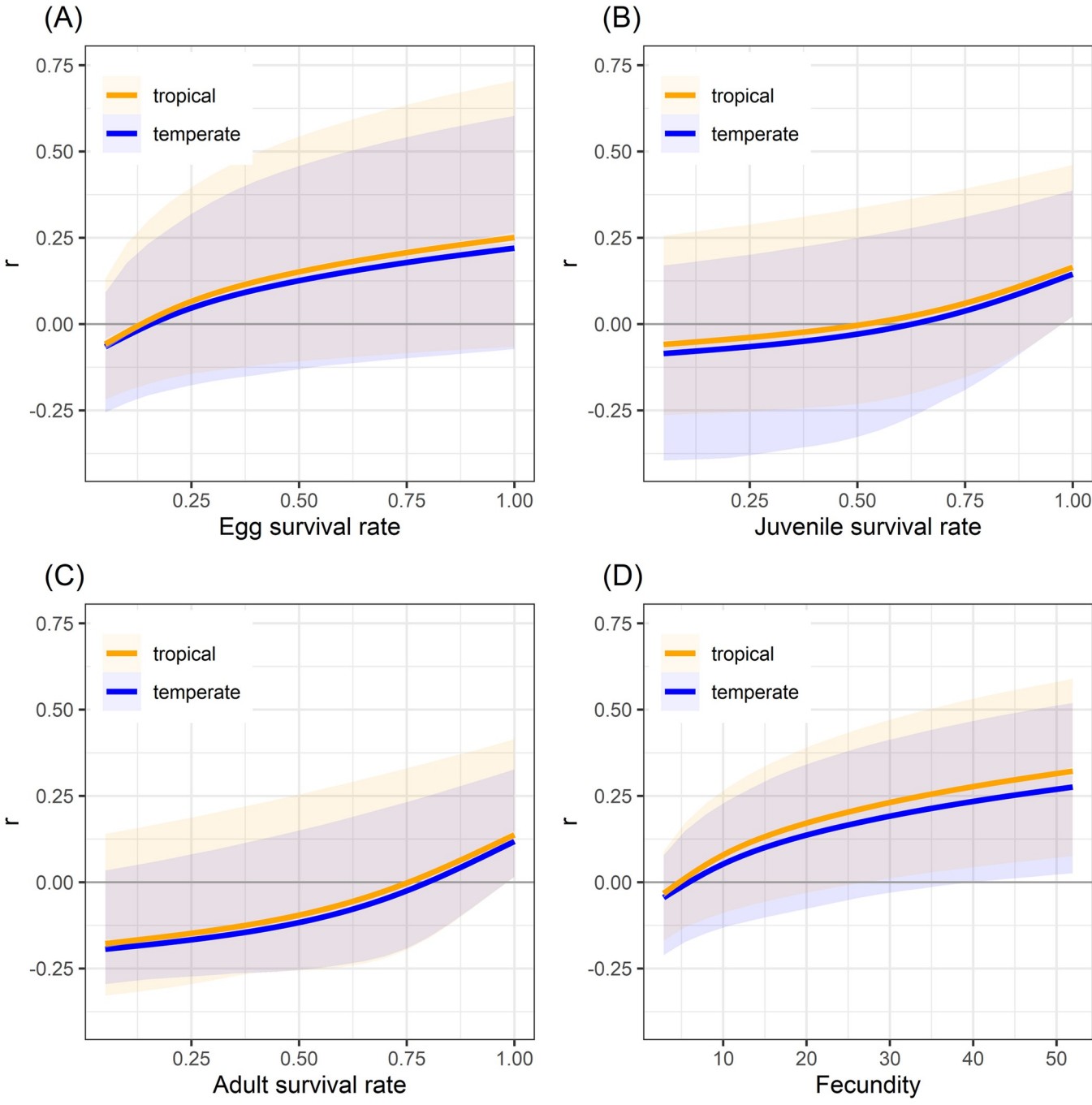

**Fig 5. Relationships between intrinsic rate of growth (*r*) and survival rates of (A) egg, (B) juvenile, and (C) adult, and (D) fecundity in freshwater turtles of tropical and temperate zones.** Population growth curves (orange and blue lines for temperate and tropical species respectively) were obtained by varying each demographic parameter with all other population parameters held constant. Confidence and prediction intervals were obtained via jackknife randomizations. White shaded polygons show 95% confidence bands. Colored polygons show 95% prediction bands (i.e. include 95% of randomized r values).

The relationship between age at sexual maturity and latitude observed in this study is also in agreement with the earlier studies [76–78]. Turtles that inhabit high latitudes reach maturity at a later age than those inhabit low latitudes. This result is likely due to more stable and more productive climate conditions at low latitudes. As growth rate in turtles depends on

**Table 5. Demographic parameters used in population modelling to estimate capacity for sustainable harvest in freshwater turtles.** Observed are median values derived from the scientific literature (S1 Table) and predicted values are from the 95% confidence set of GAM models (Table 3). "*r* min" are the minimum values necessary to obtain positive intrinsic rate of growth (*r*) as determined via sensitivity analysis (Fig 5).

| Parameter | Observed | | Predicted | | *r* min | |
|---|---|---|---|---|---|---|
| | Temp. | Trop. | Temp. | Trop. | Temp. | Trop. |
| Annual egg survival rate | 0.200 [a] | 0.200 [a] | 0.200 [a] | 0.200 [a] | 0.170 | 0.130 |
| Annual juvenile survival rate | 0.766[b] | 0.767[b] | 0.746[b] | 0.694[b] | 0.630 | 0.530 |
| Annual adult survival rate | 0.880 | 0.882 | 0.857 | 0.798 | 0.800 | 0.760 |
| Clutch size | 8.8 | 7.0 | 7.3 | 5.2 | | |
| Clutch frequency | 2.0 | 2.3 | 2.0 | 2.3 | | |
| Age at sexual maturity | 8.3 | 7.8 | 8.6 | 7.3 | | |
| Fecundity | 7.3 | 6.0 | 7.3 | 6.0 | 6.4 | 4.3 |

[a] Values derived from previous syntheses [33].

[b] Estimated as 13% less than the annual adult survival rates [62].

temperature and food availability [79, 80], thus stable warm temperature and continuous food availability in low latitudes will generate faster growth rate to reach size at sexual maturity [34]. This conclusion is also supported by the inverse relationship between Mean Temperature of Warmest Quarter and age at sexual maturity. Although it has been suggested that turtles tend to have larger body size at higher latitudes [81] a recent review (compilation of 245 species) failed to uncover clear latitudinal trends in turtle body size [39]. These differences between studies (for example [81] evaluated variation within species from a sample of 23 species of mainly northern hemisphere and temperate turtles) seem to support the hypothesis that body size latitude relationships (e.g. Bergmann's rule) maybe stronger for temperate turtle species. Large body size is thought to provide evolutionary advantages for temperate turtle species to cope with unfavorable environments e.g. via a relative increase in fasting endurance [37, 80]. As a result, temperate turtle species require longer time to reach size at sexual maturity, but increased size may provide for increased adult survivorship [33].

Adult survival and latitude were not strongly related. This was perhaps because all turtles share in common a unique morphological feature: a rigid shell [29, 82, 83]. Turtle shells not only provide physical protection from predators [29], but also important physiological functions [82–84]. The optimum benefits from the shell are achieved when a turtle has reached adult size [29] such that different environmental conditions at low and high latitudes may have little effect on adult survival rate because the shell ensures high survival regardless of ecological context.

It is important to note, however, that our failure to identify differences in survival rates may result from a lack of statistical power [55, 85]. Relatively few reports were available for survival rates of turtles at low (tropical) latitudes thereby possibly limiting the ability to detect differences might they exist. Clearly more long-term studies of turtle population biology in tropical regions are needed and would inform this analysis. This said, differences that may exist but are currently obscured by sampling variation would likely be modest and unlikely to change the overall conclusions of this study.

The distinct life-history traits of turtles at low latitudes (tropical zone) would seem to translate into greater opportunity for sustainable harvest of early stages than those at high latitudes (temperate zone, Fig 5, Table 5). However, our estimated annual sustainable harvest rate (5%) of adult turtles is considerably lower than typical thresholds for sustainable harvest rates (20%) estimated for long-lived animals [19, 86, 87]. In addition, similar to previous studies [30, 31,

35, 77, 88–91], high adult survival rates are estimated to be critical to maintain population stability due to their relatively greater contribution to population recruitment than other life stages [35]. Considering these results, harvesting wild adults would appear to present a high risk of causing population declines whether in the temperate or tropical regions, reinforcing the need to develop appropriately enforced alternate management options such as farming of captive reared turtles for meat [92].

Although adult harvest is clearly risky [9, 29, 91] there does appear to be some potential for sustainable exploitation of early stages of tropical freshwater turtle species. Indeed, egg harvest may be more feasible, because it has relatively low risk of causing population declines (Fig 5). Gibbs and Amato (29) suggest that significant additive mortality in the egg stage may not threaten population persistence and, Thorbjarnarson, Lagueux (8) identified that harvesting of eggs is the most promising strategy in the development of sustainable use programs for turtles. Sustainable use programs must of course be developed considering relevant species specific life-history traits. In the case of nest harvesting, focusing on species with sufficient reproductive output (clutch frequency, clutch size and egg mass) and ease of finding nests to be both sustainably harvested and economically viable. Integrating the conservation and harvest of eggs (for consumption, sale and/or rearing of hatchlings for the pet trade) has generated promising results for the conservation of some threatened tropical turtles e.g. *Podocnemis unifilis* in Peru [93, 94] and our analysis supports the idea that such actions could be feasible in other tropical turtle species.

We found that tropical populations could continue to grow if egg survival was reduced by up to 35%. We suggest that this surplus of eggs can be applied for both sustainable exploitation and conservation. A focus on management and sustainable exploitation of early life stages (e.g. consumption, pet trade) would also complement conservation actions that generally protect the most sensitive adult stages [9, 29]. We found that the margins for additive mortality are so tight (<10% on average in both tropical and temperate species) that the sustainable harvest of adult turtles will likely fail unless additional management actions are incorporated into conservation programs [9].

Integrated management that explicitly considers survival of all life stages is likely to generate more robust and timely increases in exploited turtle populations. Although egg survival produces a relatively small overall effect on population growth rates when compared to adult survival [29, 35], demographic simulations show that increasing survival of eggs and hatchlings can compensate for decreases in adult survival in at least one species of tropical turtle [95]. Additionally, increasing survival of early stages via community-based protection of turtle nesting beaches has been shown to provide conservation success for local communities [94], target species [94–97] and also non-target vertebrate and invertebrate taxa [96]. Whilst promising, these results come from species of the South American Podocnemididae (*P. expansa* and *P. unifilis*) that remain widely distributed and nest in areas that are both relatively accessible and easy to find for humans [98] i.e. multiple females will lay nests in the same area [94–97]. Further examples are needed to understand how the predicted surplus in early life stages can be most effectively exploited in other tropical species, especially small sized and secretive species (e.g. kinosternids in the Americas or geomydids in southeastern Asia). This understanding is required, so that populations can still increase to replace adult turtles, which remain widely targeted and threatened by additional anthropogenic impacts across tropical regions including climate change, forest loss and pollution [1, 9, 12, 18, 19].

An important caveat is that the population dynamics of temperate and tropical species in this study were evaluated using the same survival rate values for eggs due to lack of available published data on these parameters both in temperate and tropical species. Protection of egg and juvenile stages does not produce as large an effect on population growth as protecting

adult survival [29], so our conclusions are likely to remain valid despite this untested assumption. It is impossible to obtain estimates applicable to all species, but our results from stage-based population matrices provide useful reference values to analyze the relative effects of additive mortality on different stages [99, 100]. Although our choice of model maybe considered as somewhat naïve [59], with untested assumptions, they accurately represent the described life history of turtle species [58]. Future studies are needed to develop models appropriate for species specific cases. Until data are available on typical nest and juvenile survival in temperate and tropical zones, the relative impact of harvest on populations of temperate and tropical species we estimated must remain tentative.

Together the results of our study imply that sustainable harvesting is difficult to apply as a conservation strategy, both in temperate and tropical turtle species, due to the biological limitations on turtle population growth imposed by their life-history traits. This said, Eisemberg, Rose (18) suggests that complete prohibition of harvesting as a conservation strategy in turtles will not be possible to implement in tropical areas and developing countries, where local communities have long history in using turtle meat and eggs. Conservation strategies that exclude local communities in their practices are often unsuccessful at protecting wildlife [101]. Our findings support the need for sustainable harvest programs to be considered further but cautiously in the regions that have a long history of harvesting turtles for subsistence use, particularly when the species possess density dependent mechanisms to compensate harvest, such as shown in *Chelodina rugosa* [19, 43]. We reject the assumption often employed in temperate-zone turtle research that "all turtles are the same", yet also note that demographic differences we observed between temperate and tropical turtles do not translate into obviously greater opportunity for sustainable harvest of adults and juveniles in the tropics. Therefore, carefully constructed sustainable harvest programs may present greater opportunities to succeed in the tropics if based on early (egg and hatchling) stages.

## Supporting information

**S1 Table. The life-history traits data obtained from literature review.**
(DOCX)

**S1 File. Carapace length correlations.**
(DOCX)

## Acknowledgments

We thank N. E. Karraker, B. Underwood, and P. R. Sievert for discussions on ideas and their comments on this draft manuscript and to Y-H. Sung for providing life-history data for the big-headed turtle. We thank Jordi Moya-Larano, Masami Fujiwara and four anonymous reviewers for their comments on earlier versions of the text.

## Author Contributions

**Conceptualization:** Angga Rachmansah, James P. Gibbs.

**Data curation:** James P. Gibbs.

**Formal analysis:** Angga Rachmansah, Darren Norris, James P. Gibbs.

**Investigation:** Angga Rachmansah, James P. Gibbs.

**Methodology:** Angga Rachmansah, Darren Norris, James P. Gibbs.

**Resources:** Angga Rachmansah, James P. Gibbs.

**Supervision:** James P. Gibbs.

**Writing – original draft:** Angga Rachmansah, Darren Norris, James P. Gibbs.

**Writing – review & editing:** Angga Rachmansah, Darren Norris, James P. Gibbs.

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
