## [Decision Letter · Decision Letter 0]

27 Nov 2019

PONE-D-19-28843

Population Dynamics and Biological Feasibility of Sustainable Harvesting as a Conservation Strategy for Tropical and Temperate Freshwater Turtles

PLOS ONE

Dear Dr Norris,

Thank you for submitting your manuscript to PLOS ONE. After careful consideration, we feel that it has merit but does not fully meet PLOS ONE’s publication criteria as it currently stands. Therefore, we invite you to submit a revised version of the manuscript that addresses the points raised during the review process.

We received two sets of reviews. Both reviewers are relatively positive about the manuscript. One of the reviewers suggested using the phylogenetic generalized least squares (PGLS). I am not familiar with the method, but it looks promising based on my quick reading. This is an optional analysis to include if you are interested. Otherwise, it will be a potentially interesting future analysis. I also make additional comments below.

We would appreciate receiving your revised manuscript by Jan 11 2020 11:59PM. To enhance the reproducibility of your results, we recommend that if applicable you deposit your laboratory protocols in protocols.io, where a protocol can be assigned its own identifier (DOI) such that it can be cited independently in the future. For instructions see: http://journals.plos.org/plosone/s/submission-guidelines#loc-laboratory-protocols

We look forward to receiving your revised manuscript.

Kind regards,

Masami Fujiwara, PhD

Academic Editor

PLOS ONE

Journal Requirements:

3. Please ensure that you refer to Figure 3 in your text as, if accepted, production will need this reference to link the reader to the figure.

4. We note you have included a table to which you do not refer in the text of your manuscript. Please ensure that you refer to Table 1 in your text; if accepted, production will need this reference to link the reader to the Table.

Additional Editor Comments:

I note that the uses of equations 1 and 2 (lines 164 and 165) assume that the instantaneous population growth rate is close to 0 (finite population growth rate is close to 1). If it is substantially different from 0, the estimated population growth rate is biased. This is discussed in a recent paper by Kendall et al. (2019). I suggest briefly discussing the potential bias.

I also think the interpretations of the results in lines 258-271 should be done carefully. The main problem in the interpretation is that the comparison is done among relative changes in vital rates that populations can afford to bring r to 0 from the current population growth rate. All populations regardless of life-history strategies should have instantaneous population growth rate of 0 on average if they are sustainable. If it is greater than 0, there must be a reason for it. One possibility is that tropical species are severely depleted in the past and currently recovering.

It is more robust to compare the elements of sensitivity matrices (see Caswell 2001). These elements correspond to the slope of the curves plotted in Figure 5. Heppell (1998), for example, uses elasticity analysis. Because matrices are already built, the analysis should be very straightforward. The sensitivity or elasticity analyses still do not include a density-dependent process, which is the key to sustainable harvest. But Caswell et al. (2004) discuss how the sensitivity analysis is related to equilibrium density, and I also have a paper on this topic (Fujiwara 2012).

As minor comments, the interpretation part (e.g. “in aggregate the capacity for sustainable harvest of adults as an additive source of turtle mortality …” lines 270-271) should be moved to Discussion. Figure 5 was referred to in line 262, but it does not show “tropical freshwater turtle species exhibited a moderately higher intrinsic rate of growth than temperate freshwater turtle spices.” The figure only shows instantaneous population growth rate as a function of vital rates. Without knowing vital rates, we cannot tell which has a greater growth rate.

Caswell H, Takada T, Hunter CM. (2004). Sensitivity analysis of equilibrium in density-dependent matrix population models. Ecology Letters 7:380-387.

Fujiwara M. (2012). Demographic diversity and sustainable fisheries. Plos One 7:14.

Kendall BE, Fujiwara M, Diaz-Lopez J, Schneider S, Voigt J, Wiesner S. (2019). Persistent problems in the construction of matrix population models. Ecological Modelling 406:33-43.

Reviewers' comments:

Reviewer's Responses to Questions

**Comments to the Author**

1. Is the manuscript technically sound, and do the data support the conclusions?

Reviewer #1: Yes

Reviewer #2: Yes

2. Has the statistical analysis been performed appropriately and rigorously? 

Reviewer #1: Yes

Reviewer #2: Yes

3. Have the authors made all data underlying the findings in their manuscript fully available?

Reviewer #1: Yes

Reviewer #2: Yes

4. Is the manuscript presented in an intelligible fashion and written in standard English?

Reviewer #1: Yes

Reviewer #2: Yes

5. Review Comments to the Author

Reviewer #1: I notice this is the second leg of reviews and corrections. I carefully read and reviewed the response letter issued by the authors. The MS was improved from its previous version. Authors addressed all the main issues asked by the reviewers, generating a more concise MS, which is easy to read and to comprehend.

Besides the expressed above, I have some minor observations, which I include in the attached pdf file. I also think authors must address the following minor concerns from the discussion section:

1) Authors should acknowledge the importance of the reproductive effort (the amount of biomass invested in reproduction) rather the clutch size vs. egg size trade-off. Reproductive output (also estimated as relative clutch mass) is important because its analysis describe how reproductive output is packed along reproductive events.

2) The main conclusion of this MS is harvest could be feasible extracting eggs and hatchlings, nevertheless, authors should be limiting their conclusion to large species. It is almost impossible to harvest small size species such kinosternids, or very small clutch size such New World geoemydids, since their clutch size averaged 5 eggs and their nets are difficult to find. Authors also could support their harvest proposal with proved harvesting approaches such ranching or farming in crocodiles.

3) In some turtle lineages such podocnemidds could be feasible to harvest eggs and hatchlings, since these turtles nest massively in large nesting beaches and the process is conspicuous, however, authors should acknowledge that this lineage is reduced to tropical South America, and its nesting habits is more like sea turtles, however, the vast majority of freshwater and terrestrial turtles does not nest this way, but in a more secretive way.

4) Data from sea turtles nesting habits and behavior are also relevant to propose a sustainable harvest program for turtles, but, as I mentioned above, the freshwater turtles that nest collectively are reduce to only one lineage.

Reviewer #2: Major Comments

1. Why not use a PGLS to control for phylogenetic signal? Or conduct both your GAM and PGLS?

2. There are several species used in the analyses that are actually terrestrial. For example, T. carolina, T. ornata, C. mouhotti (see Bonin et al. 2006), C. flavomarginata (see Chen & Lue 1999), V. silvatica (Das 1991), M. tricarinata (Das 1991), G. spengleri (Bonin et al. 2006), H. spinosa (Das 1995; Lim et al. 1995), and G. japonica (Goris 2004) are terrestrial. Either remove these species or rephrase "freshwater" to something that reflect the dataset (e.g., predominately freshwater families).

3. How does the GAM handle highly correlated variables like body size and latitude? It appears you tested for correlations among the bioclimatic variables, so I am assuming it is important. I think you should test for correlation between body size and latitude.

Minor Comments

Abstract

Line 20 and throughout: Please replace “life history trait” with “life-history trait” throughout the manuscript.

Line 25: comma after “harvest”

Line 33: change to “stage-structured matrix”

Line 39: comma after “Yet”

Introduction

Line 60: comma after [7,8]

Line 60: Is there a more up-to-date citation for where freshwater species occur?

Methods

Line 101: How did you treat ranges if the median and mean were not given in an article for a species? In other words, what value did you extract from the range to use in the model?

Line 110: Spell out GBIF

Lines 118-121: How did you extract these variables? Did you use GIS software or google earth?

Everywhere: Figure captions were all over the place. I am assuming this wasn’t the authors’ fault.

Results

Lines 213-214: How were these 12 reports distributed between the tropics and temperate?

Fig 5 – can you include confidence intervals on these figures?

Table 2 – The font of the text is different among the columns.

Discussion

Line 316: Authors use “life-history features”, “life-history characteristics”, and “life-history traits” in this manuscript. Is there a difference among these? If not, be consist and choose one.

All other comments are embedded within the manuscript.

6. PLOS authors have the option to publish the peer review history of their article (what does this mean?). If published, this will include your full peer review and any attached files.

Reviewer #1: No

Reviewer #2: No

---

## [Author Response · Author response to Decision Letter 0]

28 Jan 2020

Below I provide our replies inline to the original email content.

De: "PLOS ONE" <em@editorialmanager.com>

Para: "Darren Norris" <dnorris75@gmail.com>

Enviado(s): 27/11/2019 12:14:23

Assunto: PLOS ONE Decision: Revision required [PONE-D-19-28843] - [EMID:5d5c1e670f9050e9]

PONE-D-19-28843

Population Dynamics and Biological Feasibility of Sustainable Harvesting as a Conservation Strategy for Tropical and Temperate Freshwater Turtles

PLOS ONE

Dear Dr Norris,

Thank you for submitting your manuscript to PLOS ONE. After careful consideration, we feel that it has merit but does not fully meet PLOS ONE’s publication criteria as it currently stands. Therefore, we invite you to submit a revised version of the manuscript that addresses the points raised during the review process.

We received two sets of reviews. Both reviewers are relatively positive about the manuscript. One of the reviewers suggested using the phylogenetic generalized least squares (PGLS). I am not familiar with the method, but it looks promising based on my quick reading. This is an optional analysis to include if you are interested. Otherwise, it will be a potentially interesting future analysis. I also make additional comments below.

Reply: We feel a more in depth phylogenetic/evolutionary analysis is beyond our manuscript aims. We justify this in replies below to reviewer 2.

We would appreciate receiving your revised manuscript by Jan 11 2020 11:59PM. To enhance the reproducibility of your results, we recommend that if applicable you deposit your laboratory protocols in protocols.io, where a protocol can be assigned its own identifier (DOI) such that it can be cited independently in the future. For instructions see: http://journals.plos.org/plosone/s/submission-guidelines#loc-laboratory-protocols

• A rebuttal letter that responds to each point raised by the academic editor and reviewer(s). This letter should be uploaded as separate file and labeled 'Response to Reviewers'.

• A marked-up copy of your manuscript that highlights changes made to the original version. This file should be uploaded as separate file and labeled 'Revised Manuscript with Track Changes'.

• An unmarked version of your revised paper without tracked changes. This file should be uploaded as separate file and labeled 'Manuscript'.

We look forward to receiving your revised manuscript.

Kind regards,

Masami Fujiwara, PhD

Academic Editor

PLOS ONE

Journal Requirements:

Reply: We have added captions and in-text citations for the Supporting Information following the plosone guidelines.

3. Please ensure that you refer to Figure 3 in your text as, if accepted, production will need this reference to link the reader to the figure.

Reply: We have added reference to Figure 3 in the Results.

4. We note you have included a table to which you do not refer in the text of your manuscript. Please ensure that you refer to Table 1 in your text; if accepted, production will need this reference to link the reader to the Table.

Reply: We have added reference to Table 1 in the Results.

Additional Editor Comments:

I note that the uses of equations 1 and 2 (lines 164 and 165) assume that the instantaneous population growth rate is close to 0 (finite population growth rate is close to 1). If it is substantially different from 0, the estimated population growth rate is biased. This is discussed in a recent paper by Kendall et al. (2019). I suggest briefly discussing the potential bias.

Reply: Thank you for sharing the reference. We have updated the text of the Methods and Discussion to include the reference and potential bias. This is a brief few sentences that we hope fairly presents the issue. We use “r” to provide a solid theoretical reference [1,2,3] to explore the potential for sustainable harvest of turtle populations. We show that early turtle life stages are most probably the best starting point for developing sustainable harvest programs in tropical turtles Although we present simulation results across the full range of possible survival values (Figure 5), our conclusions come from comparison of values within 40% of minimum positive r value (Table 4). We believe this is “close to zero” and an appropriate and robust use of “r” obtained from stable-stage population projection matrix (equations 1 and 2) that are robust and appropriate for the study group. 

[1] Sibly RM, Hone J. Population growth rate and its determinants: an overview. Philosophical Transactions of the Royal Society of London. Series B: Biological Sciences. 2002 Sep 29;357(1425):1153-70.

[2] Fuller E, Brush E, Pinsky ML. The persistence of populations facing climate shifts and harvest. Ecosphere. 2015 Sep;6(9):1-6.

[3] Pauly D, Christensen V, Guénette S, Pitcher TJ, Sumaila UR, Walters CJ, Watson R, Zeller D. Towards sustainability in world fisheries. Nature. 2002 Aug;418(6898):689.

I also think the interpretations of the results in lines 258-271 should be done carefully. The main problem in the interpretation is that the comparison is done among relative changes in vital rates that populations can afford to bring r to 0 from the current population growth rate. All populations regardless of life-history strategies should have instantaneous population growth rate of 0 on average if they are sustainable. If it is greater than 0, there must be a reason for it. One possibility is that tropical species are severely depleted in the past and currently recovering.

Reply: We show that early turtle life stages are most probably the best starting point for developing sustainable harvest programs. Our revised Discussion is duly cautious regarding the conclusions from the available data. The major limitation to further analysis is data availability as we clearly state in the Discussion (P26, L464): “An important caveat is that the population dynamics of temperate and tropical species in this study were evaluated using the same survival rate values for eggs due to lack of available published data on these parameters both in temperate and tropical species.”. The quality of the data available needs to be improved before the use of more sophisticated models can be justified.

This (instantaneous growth rate of zero) is rarely the case in the real world [1,2] with effects of harvest coupled with environmental and/or demographic stochasticity generating population cycles regardless of other ecological interactions [3,4]. The values we present are averaged across species and represent a generalization to indicate most plausible direction for future actions. We use “r” to provide a solid theoretical reference [1,2,3,4] to explore the potential for sustainable harvest of global turtle populations. We focus on sustainability of harvested populations, which are by definition not at environmental carrying capacity/equilibrium [4,5,6]. We obtain empirical values from published literature to predict and parameterize values of “Lefkovitch” matrix population models for tropical and temperate species. To identify potential starting points for sustainable harvest of tropical and temperate turtles we then analyse and present these results with reference to different life stages (Figure 5). We feel this approach is clear, robust and easily interpretable to readers. 

Obviously further data and analysis are required to establish additional metrics such as those necessary to evaluate optimal harvesting, critical population sizes, maximum growth rates for individual species [6,7,8]. Yet, we feel our conclusions are clearly supported by the analysis and provide a vital first step by identifying early stages as the most likely for sustainable harvest for subsequent analysis on freshwater turtle population dynamics and harvest sustainability.

[1] Lebreton JD. Dynamical and statistical models for exploited populations. Australian & New Zealand Journal of Statistics. 2005 Mar;47(1):49-63.

[2] Lande R, Engen S, Saether BE. Optimal harvesting of fluctuating populations with a risk of extinction. The American Naturalist. 1995 May 1;145(5):728-45.

[3] Fryxell JM, Packer C, McCann K, Solberg EJ, Sæther BE. Resource management cycles and the sustainability of harvested wildlife populations. Science. 2010 May 14;328(5980):903-6.

 [4] Sibly RM, Hone J. Population growth rate and its determinants: an overview. Philosophical Transactions of the Royal Society of London. Series B: Biological Sciences. 2002 Sep 29;357(1425):1153-70.

[5] Fuller E, Brush E, Pinsky ML. The persistence of populations facing climate shifts and harvest. Ecosphere. 2015 Sep;6(9):1-6.

[6] Pauly D, Christensen V, Guénette S, Pitcher TJ, Sumaila UR, Walters CJ, Watson R, Zeller D. Towards sustainability in world fisheries. Nature. 2002 Aug;418(6898):689.

[7] Rose KA, Cowan JH, Winemiller KO, Myers RA, Hilborn R. Compensatory density dependence in fish populations: importance, controversy, understanding and prognosis. Fish and Fisheries. 2001 Dec 1;2(4):293-327.

[8] Schaefer MB. Some considerations of population dynamics and economics in relation to the management of the commercial marine fisheries. Journal of the Fisheries Board of Canada. 1957 May 1;14(5):669-81.

It is more robust to compare the elements of sensitivity matrices (see Caswell 2001). These elements correspond to the slope of the curves plotted in Figure 5. Heppell (1998), for example, uses elasticity analysis. Because matrices are already built, the analysis should be very straightforward. The sensitivity or elasticity analyses still do not include a density-dependent process, which is the key to sustainable harvest. But Caswell et al. (2004) discuss how the sensitivity analysis is related to equilibrium density, and I also have a paper on this topic (Fujiwara 2012).

Reply: Thank you for this suggestion. We have added additional elasticity analysis to the Methods and Results. We prefer to retain Figure 5, as we believe, considering our study objectives that this is an informative way to present the results of our modelling synthesis for the diverse readership of plosone. To improve clarity for readers we have extensively revised Figure 5 following suggestions from reviewers – e.g. adding confidence and prediction bands and adding additional detail to the legend text. We are reluctant to continue further exploring changes in r / lambda (e.g. perturbation analysis), as we feel there is not sufficient data currently available to support conclusions from additional analysis. This is clearly stated in the Discussion (P26, L464): “An important caveat is that the population dynamics of temperate and tropical species in this study were evaluated using the same survival rate values for eggs due to lack of available published data on these parameters both in temperate and tropical species.”. The quality of the data available needs to be improved before the use of more sophisticated models can be justified.

As minor comments, the interpretation part (e.g. “in aggregate the capacity for sustainable harvest of adults as an additive source of turtle mortality …” lines 270-271) should be moved to Discussion. Figure 5 was referred to in line 262, but it does not show “tropical freshwater turtle species exhibited a moderately higher intrinsic rate of growth than temperate freshwater turtle spices.” The figure only shows instantaneous population growth rate as a function of vital rates. Without knowing vital rates, we cannot tell which has a greater growth rate.

Caswell H, Takada T, Hunter CM. (2004). Sensitivity analysis of equilibrium in density-dependent matrix population models. Ecology Letters 7:380-387.

Fujiwara M. (2012). Demographic diversity and sustainable fisheries. Plos One 7:14.

Kendall BE, Fujiwara M, Diaz-Lopez J, Schneider S, Voigt J, Wiesner S. (2019). Persistent problems in the construction of matrix population models. Ecological Modelling 406:33-43.

Reply: We have excluded the interpretation from the Results. To improve clarity for readers we have extensively revised Figure 5 following suggestions from reviewers – e.g. adding confidence and prediction bands and adding additional detail to the legend text. We feel this is now clarified, but are happy to follow any additional editorial guidance. 

Reviewers' comments:

Reviewer's Responses to Questions

Comments to the Author

1. Is the manuscript technically sound, and do the data support the conclusions?

Reviewer #1: Yes

Reviewer #2: Yes

2. Has the statistical analysis been performed appropriately and rigorously? 

Reviewer #1: Yes

Reviewer #2: Yes

3. Have the authors made all data underlying the findings in their manuscript fully available?

Reviewer #1: Yes

Reviewer #2: Yes

4. Is the manuscript presented in an intelligible fashion and written in standard English?

Reviewer #1: Yes

Reviewer #2: Yes

5. Review Comments to the Author

Reviewer #1: 

I notice this is the second leg of reviews and corrections. I carefully read and reviewed the response letter issued by the authors. The MS was improved from its previous version. Authors addressed all the main issues asked by the reviewers, generating a more concise MS, which is easy to read and to comprehend.

Besides the expressed above, I have some minor observations, which I include in the attached pdf file. I also think authors must address the following minor concerns from the discussion section:

1) Authors should acknowledge the importance of the reproductive effort (the amount of biomass invested in reproduction) rather the clutch size vs. egg size trade-off. Reproductive output (also estimated as relative clutch mass) is important because its analysis describe how reproductive output is packed along reproductive events.

Reply: We have followed the reviewers suggestion and clarified the Discussion as follows (P18 L343-348): “Additionally, clutch mass (number of eggs x egg size) and reproductive output (often estimated as relative clutch mass) can also vary with latitude [33, 41]. Further studies are necessary to examine how reproductive output correlates to differences in population growth rates, especially as egg size and reproductive output have been shown to be important predictors of age at sexual maturity [32, 33].”

2) The main conclusion of this MS is harvest could be feasible extracting eggs and hatchlings, nevertheless, authors should be limiting their conclusion to large species. It is almost impossible to harvest small size species such kinosternids, or very small clutch size such New World geoemydids, since their clutch size averaged 5 eggs and their nets are difficult to find. Authors also could support their harvest proposal with proved harvesting approaches such ranching or farming in crocodiles.

Reply: We have added the following to the Discussion (P22 L413 ): “Sustainable use programs must of course be developed considering relevant life-history traits. In the case of nest harvesting, focusing on species with sufficient reproductive output (clutch frequency, clutch size and egg mass) and ease of finding nests to be economically viable.”. We do not feel that it is necessary to add reference to crocodiles, which is likely to make the text more confusing for readers. We do include a highly relevant example for tropical freshwater turtles (Podocnemis species).

3) In some turtle lineages such podocnemidds could be feasible to harvest eggs and hatchlings, since these turtles nest massively in large nesting beaches and the process is conspicuous, however, authors should acknowledge that this lineage is reduced to tropical South America, and its nesting habits is more like sea turtles, however, the vast majority of freshwater and terrestrial turtles does not nest this way, but in a more secretive way.

Reply: Our objective is not to evaluate feasibility of myriad alternatives. Rather we identify an appropriate starting point for conservation action based on relevant biological parameters. There are many species (21 of the 165 in our study) with clutch sizes equal to or greater than P. unifilis. We hope future studies can build on the data presented to develop relevant case studies demonstrating the feasibility of different sustainable harvest options of early stages in different species. We have rephrased the Discussion as follows (P23, L436): “Whilst promising, these results come from two species of the South American Podocnemididae (P. expansa and P.unifilis) that remain widely distributed and nest in areas that are both relatively accessible and easy to find for humans e.g. multiple females will lay nests in the same area [91-94]. Further examples are needed to understand how the predicted surplus in early life stages can be most effectively exploited, so that populations can still increase to replace adults that remain widely targeted and threatened by additional anthropogenic impacts across tropical regions including climate change, forest loss and pollution [1, 9, 12, 18, 19].”

4) Data from sea turtles nesting habits and behavior are also relevant to propose a sustainable harvest program for turtles, but, as I mentioned above, the freshwater turtles that nest collectively are reduce to only one lineage.

Reply: Thank you for the suggestion. “collective nesting” is not necessary for sustainable harvest of eggs. Many turtles nest relatively close together, simply as a result of the availability of suitable nesting habitat. Podocnemididae have been intensively studied for over 40 years, which is why we are able to include them in the Discussion. The vast majority of tropical turtles do not have such a knowledge base. However, there are innumerable examples (e.g. North American and Australian species) where different females nest in in the same areas. We have rephrased the Discussion to clarify as follows (P23, L436): “Whilst promising, these results come from two species of the South American Podocnemididae (P. expansa and P.unifilis) that remain widely distributed and nest in areas that are both relatively accessible and easy to find for humans e.g. multiple females will lay nests in the same area [91-94]. Further examples are needed to understand how the predicted surplus in early life stages can be most effectively exploited, so that populations can still increase to replace adults that remain widely targeted and threatened by additional anthropogenic impacts across tropical regions including climate change, forest loss and pollution [1, 9, 12, 18, 19].”

Reviewer #1 Comments from .pdf file:

Abstract

This is a very long sentence, pelase reword.

Reply: We have rephrased as follows: “. Studies conducted exclusively in temperate zones have revealed that typical turtle life history traits (including delayed sexual maturity and high adult survivorship) make sustainable harvest programs an unviable strategy for turtle conservation.”

This conclusion should be adjusted to body size. Several tropical species have small sizes, withe very small eggs. The harvest of these small eggs should be for ranching only. 

Reply: We have added the following to the Discussion (P22 L413 ): “Sustainable use programs must of course be developed considering relevant life-history traits. In the case of nest harvesting, focusing on species with sufficient reproductive output (clutch frequency, clutch size and egg mass) and ease of finding nests to be economically viable.”.

Again, I think this paragraph have very long sentences. Please reword.

Reply: We have rephrased as follows: “Further studies are urgently needed to understand how the predicted population surplus in early life stages can be most effectively incorporated into conservation programs for tropical turtles.”

Numbers in the table does not seem to alligned. Pleas check the alligment of the table columns.

Reply: We have carefully revised all tables to ensure alignment, fonts and formatting are consistent and follow plosone submission guidelines.

There is an extra tab here. Reply: corrected.

Add a period here. 

Reply: We feel the sentence (P20, L369) is clear and reads well. We prefer to retain as originally submitted, but are happy to follow editorial guidance.

Change the colon for a period

Reply: We agree that this is a long sentence, yet we feel the sentence (P22, L421) is clear and reads well. We prefer to retain as originally submitted, but are happy to follow editorial guidance.

add a colon here

Reply: we believe adding a colon here (P23, L431) would be incorrect usage. We prefer to retain as originally submitted, but are happy to follow editorial guidance.

please do include the complet generic name here. It is the last sentence of the MS.

Reply: added as requested.

Reviewer #2: Major Comments

1. Why not use a PGLS to control for phylogenetic signal? Or conduct both your GAM and PGLS? Comment from .pdf file “Why not use PGLS instead of GAMs with family as a random effect?” There are plenty of turtle phylogeny out there (i.e., Pereira et al. 2017; Mol. Phylogenet Evol. 113:59-66) to collect the branch lengths. Other working with turtles (i.e., Agha et al. 2018 JEB) conducted both PGLS and GLMs and found different results.

Reply: We thank the reviewer for the interesting suggestion, but feel a more in depth phylogenetic/evolutionary analysis is beyond our manuscript aims. Looking at Table 1 and Table 2 in Agha et al. 2018 JEB (doi: 10.1111/jeb.13223) the LMM and PGLS results appear consistent. The differences reported could be attributable to other factors such as the inclusion of strongly correlated explanatory variables, i.e. Agha et. al. included both latitude and strongly correlated bioclimatic variables (e.g. mean temperature and annual precipitation) in the analysis. Table 2 is particularly revealing, as it is only the relative importance of latitude that differs in a meaningful way between LMM and PGLS results. This suggests differences could be attributable to how the different algorithms are affected by correlated variables (multicollinearity) and does not necessarily reflect any biologically meaningful difference. We have taken care to avoid collinearity between variables (all pairwise correlations <0.25). Based on these considerations, we prefer to retain our approach with Family as a random effect and believe the conclusions from our GAM analysis are robust. We hope that future studies can take advantage of the dataset we provide and explore further questions related to phylogenetic and evolutionary relationships.

2. There are several species used in the analyses that are actually terrestrial. For example, T. carolina, T. ornata, C. mouhotti (see Bonin et al. 2006), C. flavomarginata (see Chen & Lue 1999), V. silvatica (Das 1991), M. tricarinata (Das 1991), G. spengleri (Bonin et al. 2006), H. spinosa (Das 1995; Lim et al. 1995), and G. japonica (Goris 2004) are terrestrial. Either remove these species or rephrase "freshwater" to something that reflect the dataset (e.g., predominately freshwater families).

Reply: The functional traits (e.g. diet, habitat use) of different species are likely to be important for consideration in future studies. Yet, this is not a focus of our study. There is no reason to expect differences in survival between terrestrial, semi-aquatic and aquatic species. We have rephrased throughout the text and also clarified as follows in the Methods (P6, L104): “Marine turtles (Cheloniidae and Dermochelyidae) and tortoises (Testudinidae) were excluded from the results. Although some families (e.g. Emydidae) contain a mix of terrestrial (e.g. Terrapene carolina), semi-aquatic (e.g. Trachemys scripta) and aquatic (e.g. Terrapene coahuila) species, as our analysis is general across groups hereafter we refer to all as “freshwater turtles” to distinguish them from marine species or tortoises.”

3. How does the GAM handle highly correlated variables like body size and latitude? It appears you tested for correlations among the bioclimatic variables, so I am assuming it is important. I think you should test for correlation between body size and latitude.

Reply: Thank you for the interesting question. We do not agree that at the level of our study (global/species) body size is necessarily strongly correlated with latitude for our study group – freshwater turtles. We dedicated most of a paragraph to this issue (P19, L356-365), and as stated in the Discussion of our original submission (P19, L357): “Although it has been suggested that turtles tend to have larger body size at higher latitudes [77] a recent review (compilation of 245 species) failed to uncover clear latitudinal trends in turtle body size [38].” To clarify for readers we have also added the correlation test to the Methods as follows (P7, L144): “All four model variables were only weakly correlated (all pairwise correlations < 0.25) with carapace length (S1 file) so could be included in the GAM analysis [47].” We have also added Supporting information S1 file with the pairwise correlation values.

Minor Comments

Abstract

Line 20 and throughout: Please replace “life history trait” with “life-history trait” throughout the manuscript.

Reply: updated throughout the text as requested.

Line 25: comma after “harvest”. Reply: updated as requested.

Line 33: change to “stage-structured matrix”. Reply: updated as requested.

Line 39: comma after “Yet” Reply: updated as requested.

Introduction

Line 60: comma after [7,8] Reply: updated as requested.

Line 60: Is there a more up-to-date citation for where freshwater species occur?

Reply: We have also added: J. F. M. Rodrigues, M. Á. Olalla-Tárraga, J. B. Iverson, T. S. B. Akre, J. A. F. Diniz-Filho, Time and environment explain the current richness distribution of non-marine turtles worldwide. Ecography 40, 1402-1411 (2017).

Methods

Line 101: How did you treat ranges if the median and mean were not given in an article for a species? In other words, what value did you extract from the range to use in the model?

Reply: We have clarified as follows: “(midpoint calculated and used in < 1% of cases)”

Line 110: Spell out GBIF. Reply: updated as requested.

Lines 118-121: How did you extract these variables? Did you use GIS software or google earth?

Reply: we have clarified in the Methods (P7, L123) as follows: “Two bioclimatic variables relevant to freshwater turtle biology, Mean Temperature of Warmest Quarter (bio10, oC) and Precipitation of Driest Quarter (bio17, mm) were obtained from WorldClim – Global Climate Data (5-arc ≈ 10 km resolution, www.worldclim.org, [44]) and matched to the coordinates of each turtle life-history report using functions available in the raster package [45].

”

Everywhere: Figure captions were all over the place. I am assuming this wasn’t the authors’ fault.

Reply: We have included Figure captions following Plosone submission guidelines available at the time of submission (https://journals.plos.org/plosone/s/figures#loc-captions): “Place figure captions in the manuscript text in read order, immediately following the paragraph where the figure is first cited.”.

Results

Lines 213-214: How were these 12 reports distributed between the tropics and temperate?

Reply: We have clarified as follows: “Only 12 of these life history life-history trait reports (5 tropical and 3 temperate species) were from captive breeding situations while the remainder were from wild populations.”.

Fig 5 – can you include confidence intervals on these figures?

Reply: To improve clarity for readers we have extensively revised Figure 5 following suggestions from reviewers – e.g. adding confidence and prediction bands and adding additional detail to the legend text. 

Table 2 – The font of the text is different among the columns.

Reply: We have carefully revised all tables to ensure alignment, fonts and formatting are consistent and follow plosone submission guidelines.

Discussion

Line 316: Authors use “life-history features”, “life-history characteristics”, and “life-history traits” in this manuscript. Is there a difference among these? If not, be consist and choose one.

Reply: we have corrected to life-history traits throughout the text.

All other comments are embedded within the manuscript.

Below are our replies to the additional reviewer comments from the .pdf file (majority of comments in the .pdf are duplicates of the above comments):

Does this mean you had 44 species with complete data? That is to say you had 44 species with Lat, CL, CS, CF, Age, and Fecundity? If not, how did the model handle missing data?

Reply: models were built separately for each life-history trait (Methods P8 L148) and therefore had different sample sizes (as originally presented in Table 2). We have added sample sizes to Table 1. 

 What GIS software did you use to create this figure?

 Reply: Figure 2 was produced using R (ggplot2 and sf packages). We do not feel necessary to add this detail, as this is a purely visual representation and readers are free to reproduce using any of the myriad software/apps available. Following plosone guidelines we add details of the data sources in the figure legend and also provide geographic coordinates in the Supporting Information to enable others to reproduce our results.

 What about body size?

Reply: As we mention in the Methods (P8, L143) :carapace length (ln-transformed) was included to control for its well-established influence on life-history traits. We are not evaluating species level natural history. As such we do not feel that adding additional Results (and therefore Discussion) regarding body size is necessary to enable readers to evaluate the validity or robustness of our conclusions (comparison of sustainable harvest between temperate and tropical turtles). 

I don't think you need this in the manuscript. Either remove or provide in the supplemental materials.

Reply: we prefer to retain Figure 1 in the manuscript, as we feel this is relevant for the broad readership of plosone. But are happy to follow editorial guidance.

---

## [Decision Letter · Decision Letter 1]

6 Feb 2020

PONE-D-19-28843R1

Population Dynamics and Biological Feasibility of Sustainable Harvesting as a Conservation Strategy for Tropical and Temperate Freshwater Turtles

PLOS ONE

Dear Dr Norris,

Thank you for submitting your manuscript to PLOS ONE. After careful consideration, we feel that it has merit but does not fully meet PLOS ONE’s publication criteria as it currently stands. Therefore, we invite you to submit a revised version of the manuscript that addresses the points raised during the review process.

I think you have responded to all of the comments and revised the manuscript accordingly. A reviewer provided comments, and I would like to give you an opportunity to respond to them (and possibly incorporate them). Please also read the formatting and other guideline of PLoS One carefully at this stage and resubmit it.   

We would appreciate receiving your revised manuscript by Mar 22 2020 11:59PM. To enhance the reproducibility of your results, we recommend that if applicable you deposit your laboratory protocols in protocols.io, where a protocol can be assigned its own identifier (DOI) such that it can be cited independently in the future. For instructions see: http://journals.plos.org/plosone/s/submission-guidelines#loc-laboratory-protocols

We look forward to receiving your revised manuscript.

Kind regards,

Masami Fujiwara, PhD

Academic Editor

PLOS ONE

Reviewers' comments:

Reviewer's Responses to Questions

**Comments to the Author**

1. If the authors have adequately addressed your comments raised in a previous round of review and you feel that this manuscript is now acceptable for publication, you may indicate that here to bypass the “Comments to the Author” section, enter your conflict of interest statement in the “Confidential to Editor” section, and submit your "Accept" recommendation.

Reviewer #1: All comments have been addressed

2. Is the manuscript technically sound, and do the data support the conclusions?

Reviewer #1: Yes

3. Has the statistical analysis been performed appropriately and rigorously? 

Reviewer #1: Yes

4. Have the authors made all data underlying the findings in their manuscript fully available?

Reviewer #1: Yes

5. Is the manuscript presented in an intelligible fashion and written in standard English?

Reviewer #1: Yes

6. Review Comments to the Author

Reviewer #1: Authors addressed all prevouis comments. I think this is a very complete and improved version of the manuscript. It is clear that your management suggestions should be consider only for large and gregarious species. The application of your results to secretive and small size/clutch species should be wait for more data on specific study systems such emydids and kinosternids in North America or geomydids in Souteastern Asia.

7. PLOS authors have the option to publish the peer review history of their article (what does this mean?). If published, this will include your full peer review and any attached files.

Reviewer #1: No

---

## [Author Response · Author response to Decision Letter 1]

6 Feb 2020

PONE-D-19-28843R1

Population Dynamics and Biological Feasibility of Sustainable Harvesting as a Conservation Strategy for Tropical and Temperate Freshwater Turtles

PLOS ONE

Dear Dr Norris,

Thank you for submitting your manuscript to PLOS ONE. After careful consideration, we feel that it has merit but does not fully meet PLOS ONE’s publication criteria as it currently stands. Therefore, we invite you to submit a revised version of the manuscript that addresses the points raised during the review process.

I think you have responded to all of the comments and revised the manuscript accordingly. A reviewer provided comments, and I would like to give you an opportunity to respond to them (and possibly incorporate them). Please also read the formatting and other guideline of PLoS One carefully at this stage and resubmit it. 

Reply: We are glad that the Editor and reviewer agree that we have adequately responded to the extensive and detailed suggestions that followed our initial submission. We have revised the content to ensure that we follow PLoSone guideline. Based on the reviewers minor suggestion we have added a further sentence to clarify the text for readers. We have added the phrase “species specific” at line 437 : 

And extended the following sentence at line 463 to include the examples suggested by the reviewer: “Further examples are needed to understand how the predicted surplus in early life stages can be most effectively exploited in other tropical species, especially small sized and secretive species (e.g. kinosternids in the Americas or geomydids in southeastern Asia).”

We would appreciate receiving your revised manuscript by Mar 22 2020 11:59PM. To enhance the reproducibility of your results, we recommend that if applicable you deposit your laboratory protocols in protocols.io, where a protocol can be assigned its own identifier (DOI) such that it can be cited independently in the future. For instructions see: http://journals.plos.org/plosone/s/submission-guidelines#loc-laboratory-protocols

• A rebuttal letter that responds to each point raised by the academic editor and reviewer(s). This letter should be uploaded as separate file and labeled 'Response to Reviewers'.

• A marked-up copy of your manuscript that highlights changes made to the original version. This file should be uploaded as separate file and labeled 'Revised Manuscript with Track Changes'.

• An unmarked version of your revised paper without tracked changes. This file should be uploaded as separate file and labeled 'Manuscript'.

We look forward to receiving your revised manuscript.

Kind regards,

Masami Fujiwara, PhD

Academic Editor

PLOS ONE

Reviewers' comments:

Reviewer's Responses to Questions

Comments to the Author

1. If the authors have adequately addressed your comments raised in a previous round of review and you feel that this manuscript is now acceptable for publication, you may indicate that here to bypass the “Comments to the Author” section, enter your conflict of interest statement in the “Confidential to Editor” section, and submit your "Accept" recommendation.

Reviewer #1: All comments have been addressed

2. Is the manuscript technically sound, and do the data support the conclusions?

Reviewer #1: Yes

3. Has the statistical analysis been performed appropriately and rigorously? 

Reviewer #1: Yes

4. Have the authors made all data underlying the findings in their manuscript fully available?

Reviewer #1: Yes

5. Is the manuscript presented in an intelligible fashion and written in standard English?

Reviewer #1: Yes

6. Review Comments to the Author

Reviewer #1: Authors addressed all prevouis comments. I think this is a very complete and improved version of the manuscript. It is clear that your management suggestions should be consider only for large and gregarious species. The application of your results to secretive and small size/clutch species should be wait for more data on specific study systems such emydids and kinosternids in North America or geomydids in Souteastern Asia.

Reply: We are glad that the Editor and reviewer agree that we have adequately responded to the extensive and detailed suggestions that followed our initial submission. We have revised the content to ensure that we follow PLOSone guideline. Based on the reviewers minor suggestion we have added a further sentence to clarify the text for readers. We have added the phrase “species specific” at line 437 : 

And extended the following sentence at line 463 to include the examples suggested by the reviewer: “Further examples are needed to understand how the predicted surplus in early life stages can be most effectively exploited in other tropical species, especially small sized and secretive species (e.g. kinosternids in the Americas or geomydids in southeastern Asia).”

---

## [Editor Report · Decision Letter 2]

12 Feb 2020

Population Dynamics and Biological Feasibility of Sustainable Harvesting as a Conservation Strategy for Tropical and Temperate Freshwater Turtles

PONE-D-19-28843R2

Dear Dr. Norris,

We are pleased to inform you that your manuscript has been judged scientifically suitable for publication and will be formally accepted for publication once it complies with all outstanding technical requirements.

With kind regards,

Masami Fujiwara, PhD

Academic Editor

PLOS ONE
---

## [Editor Report · Acceptance letter]

13 Feb 2020

PONE-D-19-28843R2 

Population Dynamics and Biological Feasibility of Sustainable Harvesting as a Conservation Strategy for Tropical and Temperate Freshwater Turtles 

Dear Dr. Norris:

I am pleased to inform you that your manuscript has been deemed suitable for publication in PLOS ONE. Congratulations! Your manuscript is now with our production department. 

With kind regards,

on behalf of

Dr. Masami Fujiwara 

Academic Editor

PLOS ONE